# GAN-FDSR: GAN-Based Fault Detection and System Reconfiguration Method

**DOI:** 10.3390/s22145313

**Published:** 2022-07-15

**Authors:** Zihan Shen, Xiubin Zhao, Chunlei Pang, Liang Zhang

**Affiliations:** Information and Navigation School, Air Force Engineering University, Xi’an 710077, China; zh_shen520@163.com (Z.S.); zhaoxiubin926@163.com (X.Z.); zhangliang_nudt@nudt.edu.cn (L.Z.)

**Keywords:** GNSS/INS integrated system, generative adversarial networks, chaos, fault detection and identification, integrity monitoring

## Abstract

Fault detection and exclusion are essential to ensure the integrity and reliability of the tightly coupled global navigation satellite system (GNSS)/inertial navigation system (INS) integrated navigation system. A fault detection and system reconfiguration scheme based on generative adversarial networks (GAN-FDSR) for tightly coupled systems is proposed in this paper. The chaotic characteristics of pseudo-range data are analyzed, and the raw data are reconstructed in phase space to improve the learning ability of the models for non-linearity. The trained model is used to calculate generation and discrimination scores to construct fault detection functions and detection thresholds while retaining the generated data for subsequent system reconfiguration. The influence of satellites on positioning accuracy of the system under different environments is discussed, and the system reconfiguration scheme is dynamically selected by calculating the relative differential precision of positioning (RDPOP) of the faulty satellites. Simulation experiments are conducted using the field test data to assess fault detection performance and positioning accuracy. The results show that the proposed method greatly improves the detection sensitivity of the system for small-amplitude faults and gradual faults, and effectively reduces the positioning error during faults.

## 1. Introduction

The global navigation satellite system (GNSS)/inertial navigation system (INS) integrated system compensates well for the deficiencies of a single navigation system by fusing the information of GNSS and INS [1], so as to obtain more accurate and reliable navigation results, and is widely used in many fields such as agricultural automation, robot control, traffic safety, and unmanned driving [2,3,4].

Data fusion in the GNSS/INS integrated system is generally implemented using Kalman filtering algorithms [5], such as the cubature Kalman filter [6] and the unscented Kalman filter [7]. The INS is a self-contained dead-reckoning system that can accomplish navigation tasks independently of external information, which allows it to be free from environmental interference, but its errors will accumulate over time, leading to a rapid drop in positioning accuracy [8]. GNSS can provide the absolute position in real time, but its signal is fragile and susceptible to interference from many environmental factors, such as multipath effects and electromagnetic interference [9,10]. The integrated system exploits the complementary characteristics of both navigation systems, which allows it to obtain superior performance compared to a stand-alone INS or GNSS [11]. In practical navigation applications, complex observation environments often interfere with GNSS measurements. If the fault is not detected or eliminated in time, the performance advantages of integrated system disappears, and the accuracy and reliability of the system is reduced [12]. Therefore, a fault detection and system reconfiguration method that is adaptable to the environment and reliable is essential for the application of combined navigation [13].

Classical fault detection methods include the residual chi-square tests and the state chi-square tests, both of which are based on the Kalman filter [14]. They are simple to implement and detect faults quickly by judging the relationship between the distribution of the detection function and the fault threshold. Autonomous integrity monitored extrapolation (AIME) adds extrapolation cycles to the chi-square tests, and the addition of historical information enhances the detection sensitivity of the system [15]. The literature [16] combines support vector machine (SVM) with AIME to improve the ability to detect gradual faults. These methods focus on fault detection performance, and generally reconfigure the system through fault isolation in subsequent filtering. Furthermore, some methods have been proposed to reconfigure the disturbed system during fault detection. Teunissen proposed the detection, identification, and adaptation (DIA) method to detect faults and reconfigure the system [17,18], which uses the optimal model error estimation to correct the state estimation after identifying the fault to reduce the impact of the fault on the system. Modification of the classical extended Kalman filter (EKF) using robust M-estimation can effectively enhance the immunity of the system to interference [19]. The method based on the variance shift outlier model (VSOM) and EKF is able to handle single or multiple faults well in GNSS measurements by down-weighting only the identified faulty measurements while detecting faults in the raw pseudo-range data [20]. Setting two separate filters for GNSS faults and filter faults not only identifies and removes faulty measurements, but also eliminates the effects of filter faults [21]. However, the uncertainty and ambiguity of faults in complex environments will degrade the detection performance of these methods. In addition, reconfiguring the system by reducing the weight of failures alone poses a continuity risk in a low redundancy system.

With the substantial progress in artificial intelligence algorithms in recent years, neural networks are quite valuable in the field of fault detection [22]. With features such as adaptability, nonlinearity and robustness, neural networks are able to compensate for detection failures caused by traditional methods due to uncertainty and ambiguity of faults. The literature [23] combines Gaussian process regression with Kalman filtering to incorporate predicted innovation into the calculation of the fault detection function (FDF), which leads to better detection performance than with the residual chi-squared tests. The data-driven adaptive neuron fuzzy inference system (ANFIS)-based method [24] was applied to fault detection of on-board navigation sensors in unmanned aerial vehicles (UAVs) and achieved good results in terms of detection accuracy and misdetection rate. In the literature, deep neural networks have been added to the support vector machine fault detection algorithm [25]. The data fusion using the data predicted by the neural network to replace the faulty measurement when a fault occurs can effectively improve the stability of the system. Deep learning-based fault detection methods are mostly implemented based on time series prediction, and the accuracy of the prediction greatly affects the detection performance. In addition, using the difference between the current state and the predicted value to detect anomalies may be insufficient in complex and dynamic observation environments.

In recent years, generative adversarial networks (GAN) have achieved great success not only in image processing but also in time-domain sequence generation [26,27,28,29] by building deep learning models through adversarial training. Additionally, the generators and discriminators trained by adversarial training as well as the unsupervised learning technique hint at the feasibility of GAN applications in the field of fault detection [30,31]. In this paper, GAN is applied to the fault detection algorithm of a tightly-coupled GNSS/INS integration system for the first time, and a novel GAN-based fault detection and system reconfiguration method is designed which offers superior fault detection performance and more adaptable troubleshooting strategies. Unlike the method in [23,24,25] that uses the predicted values of neural networks to participate in fault detection, GAN-FDSR fully exploits the characteristics of adversarial training in the fault detection phase and designs a fault detection function based on the generation score and the discriminant score, which is more sensitive to small-amplitude faults and gradual faults. In the troubleshooting phase, a dynamic selection strategy of fault isolation and system reconfiguration is designed by analyzing the relative differential precision of positioning (RDPOP) of satellites under different observation environments, which compensates for the accuracy degradation and integrity risk brought by traditional troubleshooting methods in poorer observation environments. Furthermore, considering that the integrated navigation data are nonlinear, we analyze the chaotic characteristics of the navigation data in the data preprocessing phase, and map the one-dimensional time series to the high-dimensional space through phase space reconstruction to improve the learning efficiency of the model for nonlinear characteristics.

This paper is organized as follows. In Section 2, the data preprocessing method is presented, including the mathematical model of GNSS/INS tight integration and the reconstruction method for chaotic sequences. Section 3 describes each part of the proposed method in detail. In Section 4, simulation experiments are carried out to analyze the performance of the proposed method under different environments. Finally, the summary for this paper is presented in Section 5.

## 2. Data Preprocessing

### 2.1. Tightly Coupled GNSS/INS Integration Model

In the tightly coupled GNSS/INS integration model, the INS module solves the navigation parameters separately, and then performs integrated filtering with the GNSS measurements, and the obtained estimated error is used to correct the INS output. The specific flow chart is shown in Figure 1.

Tightly coupled GNSS/INS integration contains two parts: the state model and the measurement model. The state model consists of the error state equations of INS and GNSS, which is usually expressed as:(1)$$\dot{X}=FX+GW$$
where F is the system dynamic matrix, G is the noise coefficient matrix, and W is the system noise vector.

The state vector in the filter is given by:(2)$$X={\left[{\left(\varepsilon \right)}^{T},{\left(\delta V\right)}^{T},{\left(\delta P\right)}^{T},{\left(\delta w_{ib}\right)}^{T},{\left(\delta f\right)}^{T},\delta t_{u},\delta t_{ru}\right]}^{T}$$
where ε, δV, and δP denote the INS error states of attitude, velocity, and position, respectively; $\delta w_{ib}$ and δf represent the gyro and accelerometer bias vectors; $\delta t_{u}$ and $\delta t_{ru}$ denote the range bias and range drift related to the receiver clock, respectively.

The measurement model can be written as:(3)Z=HX+V
where Z, H and V donate the measurement vector, the measurement matrix and the measurement noise vector, respectively.

Assuming a discrete-time process, the state equation and measurement equation can be rewritten as follows:(4)$$X_{k}=\Phi_{k,k-1}X_{k-1}+\Gamma_{k,k-1}W_{k-1}$$
(5)$$Z_{k}=H_{k}X_{k}+V_{k}$$
where $X_{k}$, $\Phi_{k,k-1}$, $Z_{k}$ and $H_{k}$ are the state vector, transition matrix, the measurement vector and the measurement model matrix, respectively. $W_{k-1}$ and $V_{k}$ are the process noise and measurement noise, and both are zero-mean Gaussian white noise with process covariance matrices $Q_{k}$ and $R_{k}$.

State update in extended Kalman filtering consists of state estimation and one-step prediction of the covariance matrix, which can be given as follows:(6)$${\bar{X}}_{k}=\Phi_{k,k-1}{\hat{X}}_{k-1}$$
(7)$$P_{k,k-1}=\Phi_{k,k-1}P_{k-1}\Phi_{k,k-1}^{T}+\Gamma_{k-1}Q_{k-1}\Gamma_{k-1}^{T}$$
where $\hat{X}$ is the estimation of the state vector and $P_{k,k-1}$ is its covariance matrix.

The measurement update in the filter is given by:(8)$${\hat{X}}_{k}={\bar{X}}_{k,k-1}+K_{k}v_{k}$$
(9)$$P_{k}=(I-K_{k}H_{k})P_{k,k-1}$$
where $K_{k}$ is the Kalman gain matrix, $v_{k}$ is the residual vector, I denotes the identity matrix. $K_{k}$ and $v_{k}$ can be rewritten as follows:(10)$$K_{k}=P_{k,k-1}H_{k}^{T}{\left(H_{k}P_{k,k-1}H_{k}^{T}+R_{k}\right)}^{-1}$$
(11)$$v_{k}=Z_{k}-H_{k}{\bar{X}}_{k,k-1}$$

### 2.2. Chaotic Characteristic Analysis and Phase Space Reconstruction

There are many factors that affect time series analysis, especially in dynamical systems where the dynamical equations of these factors are often nonlinear and more likely to be chaotic. If a sequence shows chaotic properties, reconstructing it in phase space is an important method to study the state of a dynamical system. The essence of this approach is to transform the extension of the time series into the interpolation of phase space. Based on the Takens’ embedding theorem [32], the optimized delay time and minimum embedding dimension are determined according to the actual time series. The one-dimensional time series can then be reconstructed into a phase space with the same topological meaning as the motive power system. A reasonable selection of the delay time and embedding dimension then allows reconstruction of the phase space with satisfactory adaptability, thereby producing the internal nonlinear mapping. By exploiting the properties of the Lyapunov exponent, this approach can recognize and make short-term predictions of chaotic time series [33,34].

The time delay and the embedding dimension are the keys to phase space reconstruc-tion. However, there are no uniform selection criteria [35]. In this section, we will analyze the chaotic characteristics of navigation data, and use the mutual information method [36] and CAO method [37] to determine these parameters. Concerning the latter method, CAO is the last name of Professor Cao Liangyue, who proposed the method that can solve the embedding dimension with a small amount of data.

Here, $\rho_{G}^{k}(\varepsilon )$ and $\rho_{I}^{k}(\varepsilon )$ are defined as the GNSS pseudo-range and the INS pseudo-range of the kth satellite. The difference in the pseudo-range increment for the kth satellite can be calculated as:(12)$$\rho_{D}^{k}(\varepsilon )=[\rho_{G}^{k}(\varepsilon +1)-\rho_{G}^{k}(\varepsilon )]-[\rho_{I}^{k}(\varepsilon +1)-\rho_{I}^{k}(\varepsilon )]$$
where ε=1,2,3,⋯,N. A time series $R=\{r_{1},r_{2},r_{3},\cdots ,r_{N}\}$ is obtained, where $r_{N}=\rho_{D}^{k}(N)$.

Next, we take the data of the 8th satellite as an example to analyze the chaotic characteristics of R. The details of the experimental data are presented in Section 4.

#### 2.2.1. Determining the Time Delay

The mutual information method is a well-established method for determining the delay time which can describe the nonlinearity of the sequence and maximize the amount of information contained in the time series.

We expand the one-dimensional time series R to the two-dimensional phase space (A,B), and equate the phase space into β×β cells, where $A=\{r_{1},r_{2},r_{3},\cdots ,r_{N-\tau }\}$, $B=\{r_{\tau +1},r_{2},r_{3},\cdots ,r_{N}\}$, and the value interval of β is [100, 200] generally. The following formula can be obtained:(13)$$E(\tau )=-\sum_{ab}p_{ab}(\tau )\ln\frac{p_{ab}(\tau )}{p_{a}p_{b}},\;a,b=1,\cdots ,\beta$$
where E(τ) is the average information entropy of the time series; $p_{a}=\frac{Row[a]}{N-\tau }$ denotes the probability that the phase point of (A,B) falls in the ath column cell; $p_{b}=\frac{Col[b]}{N-\tau }$ denotes the probability that the phase point falls in the bth row cell; Row[a] and Col[b] denote the counts of phase points in row a and column b, respectively; $p_{ab}(\tau )$ is the joint probability; and τ is the time delay. As τ increases, E(τ) of $r_{t}$ and $r_{t+\tau }$ gradually decrease until a critical point where they are no longer correlated. This point is usually taken as the first minimal point.

Figure 2 shows the average information entropy at different delay times. At 10 s, the average information entropy reaches a minimal value for the first time, which indicates that the correlation between sequence fragments reaches a minimum at this time, so τ is 10 s.

#### 2.2.2. Finding the Embedding Dimension

Using the CAO method to find the embedding dimension enables the reconstructed time series to be free of information folding in the phase space, which is widely used in chaotic sequence analysis. The algorithm steps are as follows.

Starting from a smaller embedding dimension m, the m-dimensional and m+1-dimensional phase spaces are reconstructed. The ratio of Euclidean distances between adjacent points in phase space h(i,m) is calculated as follows:(14)$$h(i,m)=\frac{{\left|\left|r_{i}^{m+1}-r_{\hat{i}}^{m+1}\right|\right|}_{\infty }}{{\left|\left|r_{i}^{m}-r_{\hat{i}}^{m}\right|\right|}_{\infty }}$$
where $r_{i}^{m+1}$ and $r_{\hat{i}}^{m+1}$ represent the ith and $\hat{i}\mathrm{th}$ state points in the m+1-dimensional phase space; $r_{\hat{i}}^{m}$ is the vector with the smallest Euclidean distance from $r_{i}^{m}$ in the m-dimensional phase space, $\hat{i}=1,2,\cdots ,M$, where M=N−(m−1)τ and $\hat{i}\neq i$; ${\left|\left|r_{i}^{m}-r_{\hat{i}}^{m}\right|\right|}_{\infty }$ represents the Euclidean distance between $r_{i}^{m}$ and $r_{\hat{i}}^{m}$.

The average of all ratios in m-dimensions is calculated as:(15)$$E(m)=\frac{1}{N-1-m\tau }\sum_{i=1}^{N-1-m\tau }h(i,m)$$

We then define $E_{m}=\frac{E(m+1)}{E(m)}$ to describe the change in E(m) when the embedding dimension is changed, and repeat the above steps until $E_{m}=\frac{E(m+1)}{E(m)}$ is stable.

When m increases to a certain value, $E_{m}$ converges to 1 if the sequence has chaotic properties. At this point the attractor of the time series has been completely opened in the phase space, and the increase of the embedding dimension will no longer make the signature sequence contain more information. The variation of $E_{m}$ in each dimension according to the CAO method is shown in Figure 3. When the dimension reaches 9, the variation is dynamically stable in a small range. At this point, it can be considered that the reconstruction of the time series in this dimension can reflect the change pattern of the original system approximately in the phase space, so the embedding dimension is 9.

#### 2.2.3. Calculating the Lyapunov Exponent

The maximum Lyapunov exponent is an important index to judge whether the signal has chaotic characteristics. If the Lyapunov exponent is greater than 0, it means that no matter how small the initial distance between the two trajectories is in the phase space of the system, the difference will increase exponentially with time and eventually chaos will emerge. The steps for calculating the Lyapunov exponent are as follows:

1.Perform a fast Fourier transform (FFT) on the time series to obtain the average period P;

2.Reconstruct R in the phase space according to m and τ. The obtained multidimensional sequence can be expressed as:(16)$$Y=\left[\begin{array}{l} Y_{1}\\ Y_{2}\\ \vdots \\ Y_{M} \end{array}\right]=\left[\begin{array}{ccccc} r_{1} & r_{1+\tau } & r_{1+2\tau } & \cdots & r_{1+(m-1)\tau }\\ r_{2} & r_{2+\tau } & r_{2+2\tau } & \cdots & r_{2+(m-1)\tau }\\ \vdots & \vdots & \vdots & \vdots & \vdots \\ r_{M} & r_{M+\tau } & r_{M+2\tau } & \cdots & r_{M+(m-1)\tau } \end{array}\right]$$
where Y is an M×m dimensional matrix, $Y_{1}$ denotes the phase points, and M is the number of phase points in the reconstructed phase space;

3.Find the closest point $Y_{\hat{j}}$ for each $Y_{j}$ in the phase space and calculate the initial distance as follows:(17)$$d_{j}(0)=\underset{}{\min}\left\|Y_{j}-Y_{\hat{j}}\right\|,\;\left|j-\hat{j}\right|>P$$
where $\hat{j}=1,2,\cdots ,M$ and $\hat{j}\neq j$;

4.Calculate the distance between each $Y_{j}$ and the corresponding $Y_{\hat{j}}$ in the phase space after i steps.
(18)$$d_{j}(i)=\left\|Y_{j+i}-Y_{\hat{j}+i}\right\|,\;t=1,2,\cdots ,\min(m-j,M-\hat{j})$$

5.Calculate the average of all $\ln d_{j}(i)$ for each i as follows:(19)$$D(t)=\frac{1}{q\Delta t}\sum_{j=1}^{q}\ln d_{j}(i)$$
where q is the number of non-zero $d_{j}(i)$, and Δt represents the sample period. The slope of the line is the largest Lyapunov exponent when linear regression is performed by the least square method.

Table 1 lists the chaos indices of the visible satellites. Clearly, the time series R of each visible satellite has chaotic characteristics in the experiment, and the data can be reconstructed using chaos theory.

## 3. GAN-FDSR: GAN-Based Fault Detection and System Reconfiguration Method

### 3.1. Model Training

The left part of Figure 4 shows the training structure of the model. In this work, the popular variants of the recurrent neural network (RNN) named gated recurrent unit (GRU) networks were employed to build base models of the generator and discriminator to capture temporal correlations of navigation data [38]. Similar to the long short-term memory neural network (LSTM), GRU solves the problems of gradient disappearance, gradient explosion and a lack of long-term memory capacity of RNN, allowing RNN to utilize long-range temporal information effectively. [39]. In addition, GRU models have fewer parameters and less complicated structures compared with LSTM models so it requires less time for model training [40]. The generator takes a random sequence in the latent space as input to generate a pseudo-sequence that is similar to the original sequence. The discriminator takes the original sequence and the pseudo-sequence generated by the generator as input, and judges the authenticity of the data. During training, the generator aims to generate pseudo-sequences that can deceive the discriminator, while the discriminator aims to recognize the pseudo-sequences. The generator and discriminator compete with each other and co-evolve to eventually reach a dynamic equilibrium.

In adversarial training, we consider the sequences of the training set to be normal, while the pseudo-sequences produced by the generator are abnormal. Before the training, the dataset needs to be partitioned into multiple sub-sequences to achieve better training results. We partition Y into a set of subsequences $Y^{test}=\{y_{s},s=1,2,\cdots ,n\}$ using a sliding window of window size w and step size u, where $n=\frac{\left(M-w\right)}{u}$. In addition, a set of multi-dimensional sequences $Z=\{z_{s},s=1,2,\cdots ,n\}$ with the same dimension as $Y^{test}$ is taken from the random space as input to the generator. The model training strategy is as follows.

Feed random sequences generated from random space into the generator to generate pseudo-sequences.
$$Z=\{z_{s},s=1,2,\cdots ,n\}\Rightarrow G(Z)$$

Pseudo-sequences and original sequences are inputted to the discriminator for discriminating.
$$Y^{test}=\{y_{s},s=1,2,\cdots ,n\}\Rightarrow D(Y^{test})\;G(Z)\Rightarrow D(G(Z))$$

Optimize the parameters of the discriminator.
$$\min\frac{1}{n}\sum_{s=1}^{n}\left[-\log D(x_{s}\right)-\log(1-D(G(z_{s})))]$$

Optimize the parameters of the generator.
$$\min\sum_{s=1}^{n}\log(-D(G(z_{s})))$$

Record the parameters of the discriminator and generator in the current iteration after completing the iterative training.

### 3.2. Detection Function and Detection Threshold

After sufficient iterations of training, the pseudo-sequence is extremely similar to the original sequence, and the discriminator is sufficiently sensitive to anomalous sequences, although it is difficult to distinguish the pseudo-sequence from the original sequence. Therefore, the fault detection function consists of two parts: the generation score and the discrimination score.

Supposing that the system obtains the pseudo-range at the moment N+2, we can derive $r_{N+1}$ according to Equation (12) and reconstruct it in phase space using Equation (16). Eventually a multi-dimensional sequence $y^{new}$ of m dimensions and length w can be obtained, where $r_{N}=y_{w,m}^{new}$.

The discriminator score can be obtained from $D(y^{new})$ directly, but computing the generator score requires the pseudo-sequence that is most similar to the input sequence. In other words, the random sequence that best matches the $y^{new}$ needs to be found. First, $z^{1}$ is extracted from the random space and fed into the generator to obtain a pseudo-sequence $G(z^{1})$. Then we use the generation error obtained by calculating the similarity between $G(z^{1})$ and $z^{1}$ as the gradient to update z.
(20)$$\underset{z^{k}}{\min}\Lambda (y^{new},G(z^{k}))=1-Cov(y^{new},G(z^{k}))$$

After sufficient iterations, the $z^{k}$ that makes $G(z^{k})$ most similar to $y^{new}$ is the random sequence corresponding to $y^{new}$. The generation scores is calculated as follows:(21)$$Gs(y^{new})=\left|y^{new}-G(z^{k})\right|$$

The fault detection function for $y^{new}$ can be written as:(22)$$Score(y^{new})=\lambda Gs(y^{new})+(1-\lambda )D(y^{new})$$
where λ is the weight associated with the training effect of the generator and the discriminator. The performance of the generator and discriminator reaches a dynamic equilibrium after the training is completed. λ can be calculated according to the dispersion degree of the generation and discrimination errors. The specific calculation process is as follows.

1.Calculate the generation error using the pseudo-sequences and the original sequences: $R^{G}=\left\{\left|y_{s}-G(z_{s})\right|,s=1,2,\cdots ,n\right\}$;

2.$L^{G}=\left\{l_{s}^{G},s=1,2,\cdots ,n\right\}$ is obtained after normalization, where $l_{s}^{G}=\frac{\left|y_{s}-G(z_{s})\right|-\min(R^{G})}{\max(R^{G})-\min(R^{G})}$.

3.The dispersion degree of the generation error can be given as:(23)$$S^{G}=\sqrt{\frac{\sum_{s=1}^{n}{\left(l_{s}^{G}-{\bar{l}}^{G}\right)}^{2}}{n-1}}$$
where ${\bar{l}}^{G}$ is the mean value of the elements in $L^{G}$.

The classification accuracy $\left\{c_{u},u=1,2,\cdots ,2n\right\}$ is obtained by feeding $Y^{test}$ and pseudo-sequences into the trained discriminator, and the discriminant error can be expressed as $R^{D}=\left\{\left|c_{u}-0.5\right|,u=1,2,\cdots ,2n\right\}$. Similarly, the dispersion degree of the discriminant error $S^{D}$ can be calculated by normalizing and calculating the standard deviation. The weight can be expressed as:(24)$$\lambda =\frac{S^{D}}{S^{G}+S^{D}}$$

If the navigation system is fault-free, then there is a null hypothesis $H_{0}:Score\sim N(\mu_{1},\sigma )$, otherwise there is an alternative hypothesis $H_{1}:Score\sim N(\mu_{2},\sigma )$. In other words, the mean value of Score is shifted when a fault occurs. We can construct the following detection criteria through the tolerable false alarm rate α:(25)$$\left\{\begin{array}{l} Score>T2,\;\mathrm{fault}\;\mathrm{occurs}\\ Score\leq T2,\;\mathrm{no}\;\mathrm{fault}\;\mathrm{occurs} \end{array}\right.$$
where T2 is the detection threshold of the false alarm rate α.

### 3.3. System Reconfiguration

When the observation redundancy is high, isolating fault measurements introduces small positioning errors because the system has high integrated filtering accuracy. However, when the satellite configuration is poor or the number of visible satellites is low, isolating fault observations reduces the available information, which leads to a rapid decrease in filtering accuracy with increasing fault duration, and eventually seriously affects the accuracy and reliability of the whole system.

In practical navigation applications such as UAV and vehicle navigation, the system needs to be able to adjust the reconfiguration scheme dynamically when the satellite observation environment becomes unstable due to factors such as building block and electromagnetic interference. The influence on the integrated filtering is investigated by calculating the RDPOP of the faulty satellite. The formula is as follows:(26)$$\mathrm{RDPOP}=\frac{\sqrt{\sum_{j=7}^{9}{\tilde{P}}_{ki}^{jj}}-\sqrt{\sum_{j=7}^{9}P_{k}{}^{jj}}}{\sqrt{\sum_{j=7}^{9}P_{k}{}^{jj}}},\;i=1,\cdots ,n$$
where i denotes the dimension of the isolated observation; ${\tilde{P}}_{ki}^{jj}$ and $P_{k}{}^{jj}$ represent the jth element on the diagonal of the covariance matrix ${\tilde{P}}_{ki}$ and $P_{k}$, respectively; ${\tilde{P}}_{ki}$ is the covariance matrix of new state estimation after fault isolation consistent with Equation (A6) in Appendix B; and $P_{k}$ is consistent with Equation (9).

RDPOP reflects the decrease in positioning accuracy after isolating a satellite. Therefore, a system reconfiguration scheme can be constructed based on RDPOP as follows:(27)$$\left\{\begin{array}{l} \mathrm{RDPOP}>\eta ,\;\mathrm{measurement}\;\mathrm{reconfiguration}\\ \mathrm{RDPOP}\leq \eta ,\;\mathrm{fault}\;\mathrm{isolation} \end{array}\right.$$
where η is the detection threshold of RDPOP, which is determined according to the requirements of filtering accuracy generally. The fault measurement is isolated when the RDPOP of the fault satellite is less than η. Otherwise, the pseudo-range calculated by $G(z^{k})$ and Equation (12) replace the faulty measurement to integrated filtering.

The specific process of the GAN-based fault detection and system reconfiguration method is as follows.

①Preprocess raw pseudo-range data.

②The pre-processed data is fed into a trained GAN model to calculate the detection score. The system is considered faulty and requires system reconfiguration if the score is greater than the detection threshold T2, otherwise the integrated filtering continues.

③Calculate the RDPOP value of the faulty satellite. If RDPOP is greater than η, use the generated pseudo-range to replace the faulty measurement for integrated filtering; otherwise, isolate the fault measurement. 

④Obtain the navigation solution of the tightly coupled system by correcting the INS output with the error estimation from the integrated filtering.

The architecture of GAN-FDSR is shown in Figure 5.

## 4. Field Test Results and Analysis

The actual data is collected by a vehicular integrated navigation system which consists of a vehicle, two GNSS antennas, and a POS320 inertial navigation device. The inertial sensor specifications of the POS320 are listed in Table 2. The test equipment is shown in Figure 6.

### 4.1. Analysis of RDPOP

In order to analyze the RDPOP of satellites under different environments and the influence on positioning accuracy, four groups of satellite observation environments provided in Table 3 are set artificially based on the measured data, and the positioning deviation of the faulty satellite is 20 m.

The RDPOP of the faulty satellite in each environment is shown in Figure 7, with η set to 0.1. The system reconfiguration is performed using the fault isolation (FI) and fault adaptation (FA) described in detail in Appendix B and Appendix C, respectively. The positioning errors in each environment during the fault are shown in Figure 8.

In Figure 8, the positioning errors of Env 1 and Env 3 are much smaller than those of Env 2 and Env 4. Although the faulty satellite of Env 1 and Env 2 are the same, the positioning error of Env 2 increases sharply because of the decrease in the number of visible satellites. Env3 and Env4 both have five visible satellites, but the positioning error varies greatly. This is because G11 has a greater effect on the positioning accuracy under configuration 2, which leads to a large increase in the positioning error.

Figure 7 and Figure 8 reveal that the faulty satellites with large RDPOP pose enormous challenges for system reconfiguration. FI and FA introduce substantial loss of positioning accuracy in poor observation environments. Therefore, the system reconfiguration scheme should be able to adjust to the changes of the observation environment dynamically according to the weight of the faulty satellite in the filter. 

### 4.2. Analysis of Fault Detection Performance and Positioning Accuracy

In order to verify the advantages of the method proposed in this paper, a fault is added to the measurement of G7 during 300–360 s. The configuration of the satellites and the RDPOP of G7 are shown in Figure 9. Two additional methods are introduced for comparison. One is the fault detection and isolation method (FDI), and the other is the fault detection and adaptation method (FDA). The fault detection methods for FDI and FDA are given in Appendix A, and their system reconfiguration methods are FI and FA respectively.

The types of faults in the integrated navigation can be roughly divided into large-amplitude faults, small-amplitude faults and gradual faults. Large-amplitude faults are simple to detect due to rapid and large changes, whereas small-amplitude faults and gradual faults are a problem in fault detection due to their small initial amplitude. To test the GAN-FDSR fault detection performance, five fault types are set which contain two step faults and three gradual faults. The specific fault parameters are shown in Table 4. The false alarm rate is 0.001 in the experiment. The detection threshold for FDI and FDA is 2.88, denoted as T1, according to Appendix A. The detection threshold for GAN-FDSR is 1.2, denoted as T2.

Figure 10 shows the fault detection results of the three methods in each environment. Large-amplitude faults lead to rapid changes in the detection function, so all the methods perform well under fault Type 1. It is worth noting that the detection function of FDI and DIA decreases gradually after the fault is identified, which is caused by the filtering error. The detection function changes rapidly due to the large-amplitude fault, so all methods perform well under Type 1. The step fault in Type 2 has a small amplitude which leads to the failure of FDI and FDA to identify the fault, but GAN-FDSR performs well in detecting this fault. In addition, the detection delay occurs in all methods when gradual faults occur, and GAN-FDSR has much smaller detection delays compared with FDI and FDA.

Table 5 lists the detection delay and missing alarm rate of three methods to demonstrate the detection performance. Both FDI and FDA use the residual chi-square test in the experiment, so their detection performance is similar. Benefiting from the sensitivity to small-amplitude faults, GAN-FDSR has lower detection delay and missing alarm rate compared to traditional methods, which brings huge performance advantages.

In addition to fault detection performance, positioning accuracy is another critical metric in navigation. In Figure 9, it can be seen that G7 has a large impact on the integrated filtering during a fault, which creates a large troubleshooting challenge. Therefore, improper troubleshooting may lead to a rapid accumulation of positioning errors which brings great risks to users.

Figure 11 gives the positioning errors of the three methods for different fault types, where FF represents fault-free. The positioning errors of FDI and FDA show a cumulative tendency over time except for the step-type positioning errors caused by undetected faults under Type 2. Although FDI and FDA reduce the localization error after the fault is identified, the absence of valid measurements increases the positioning error again over time. In contrast, GAN maintains a lower positioning error with a smaller error accumulation rate under all types of faults. Considering both Figure 11 and Figure 9, we observe that GAN-FDSR adjusts the system reconfiguration strategy in time when the RDPOP exceeds the threshold.

Considering that there will be a certain amount of error even if the system is fault-free, Figure 12 shows the errors of the three methods under each fault type relative to fault-free system to analyze the positioning accuracy quantitatively. Although the fault in Type 1 was identified, the positioning errors of FDI and FDA were the largest among all the types, which demonstrates once more that caution is necessary in handling fault observations when the observation environment is poor. However, the errors of GAN-FDSR are quite close under each type and much lower than the other methods as the integrated system benefits from the timely reconfiguration of the measurements and the small error in the generated data.

## 5. Conclusions

In this paper, a GAN-based fault detection and system reconfiguration method for a tightly coupled GNSS/INS integrated system is proposed. This method greatly improves the fault detection performance and positioning accuracy, and helps to improve the integrity and reliability of the system. A novel fault detection method is designed based on the generator and discriminator of GAN, which departs from the traditional method that using residuals for fault detection. We investigated the feasibility of chaos theory for the processing of pseudo-range data to improve the learning efficiency of the model for nonlinear characteristics through phase space reconstruction. We also analyzed the relationship between RDPOP and positioning accuracy under different observation environments to develop a dynamic system reconfiguration scheme. Finally, several experiments were conducted to assess fault detection performance and positioning accuracy using measured data. The results show that: (a) The method shows excellent detection performance and significantly reduces detection delays and missing alarm rates for difficult small-amplitude and gradual faults. (b) The positioning error of the system during faults is reduced effectively by dynamically selecting an appropriate system reconfiguration strategy according to different environments.

However, the study presented in this paper is based on the assumption that the INS always works properly, and the possibility of inertial sensor failures needs to be taken into account to improve the method in the future. In addition, the incipient fault diagnosis of the integrated navigation system also deserves further study [41].

## Figures and Tables

**Figure 1 sensors-22-05313-f001:**
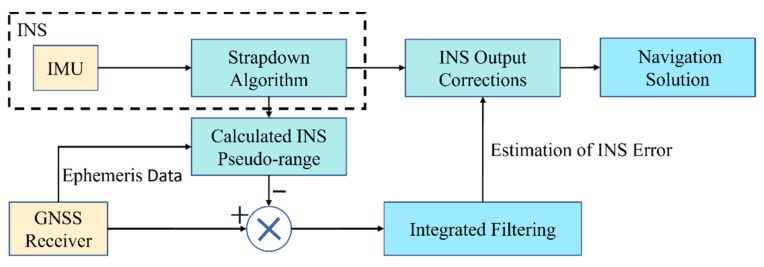
GNSS/INS tight integration architecture.

**Figure 2 sensors-22-05313-f002:**
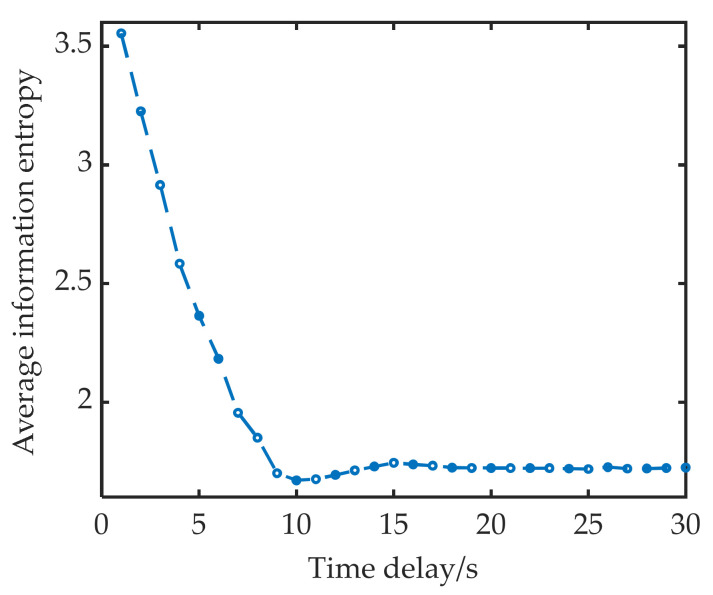
Average information entropy under different delay times.

**Figure 3 sensors-22-05313-f003:**
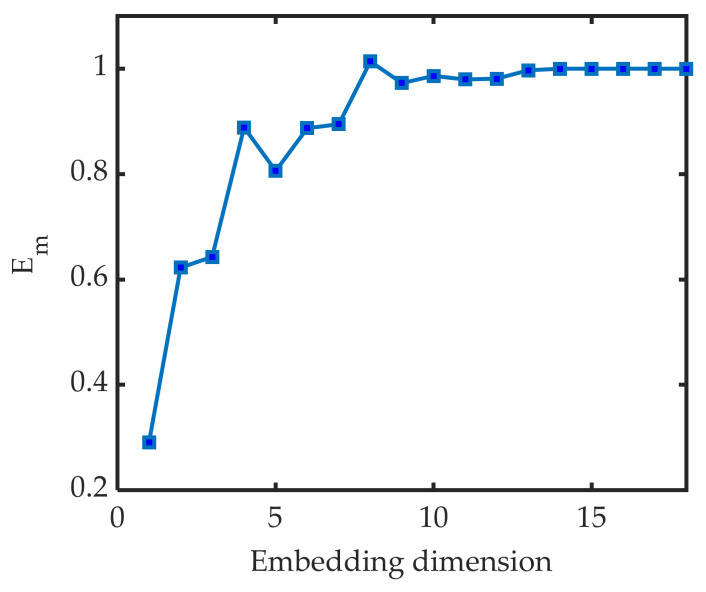
Using CAO method to find the embedding dimension.

**Figure 4 sensors-22-05313-f004:**
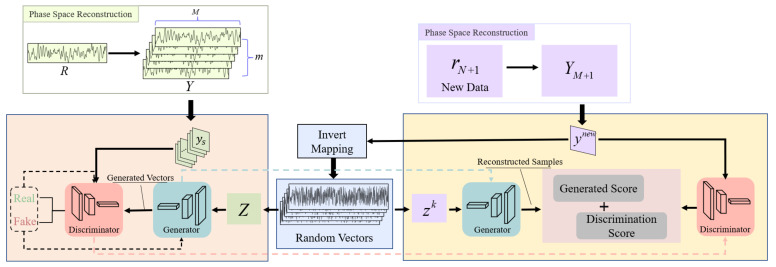
Model training and fault detection score generation. On the left is shown the training process of the model, and on the right is the process of calculating the fault detection score.

**Figure 5 sensors-22-05313-f005:**
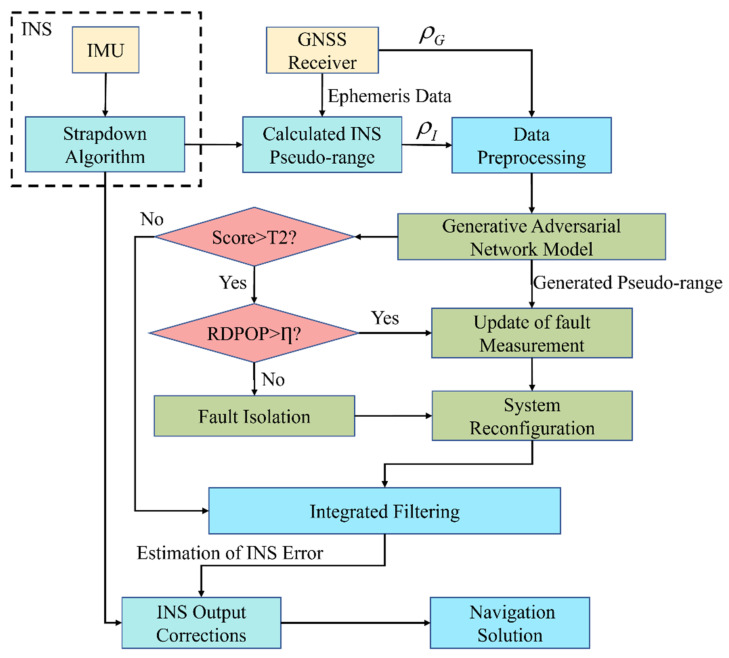
The architecture of GAN-FDSR.

**Figure 6 sensors-22-05313-f006:**
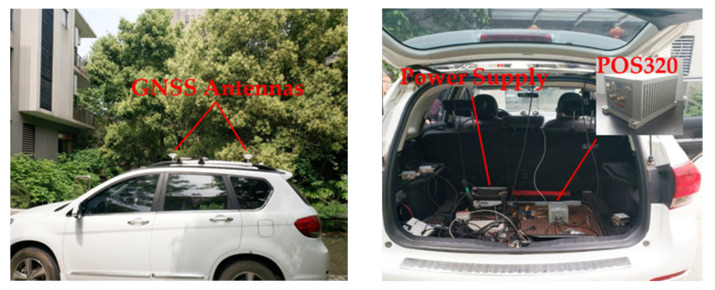
Test equipment.

**Figure 7 sensors-22-05313-f007:**
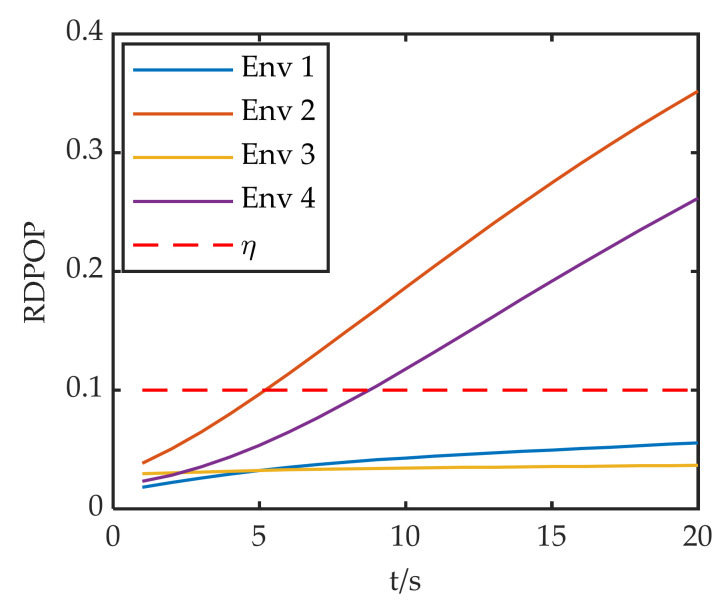
RDPOP of the faulty satellite in each environment.

**Figure 8 sensors-22-05313-f008:**
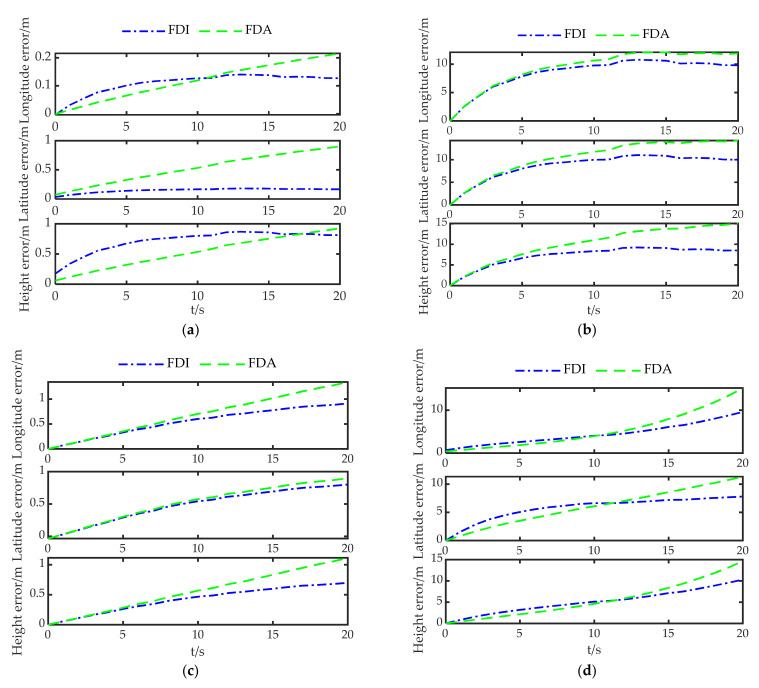
Positioning errors in each environment during the fault. (**a**–**d**) are the positioning errors under Env1, Env2, Env3 and Env4, respectively.

**Figure 9 sensors-22-05313-f009:**
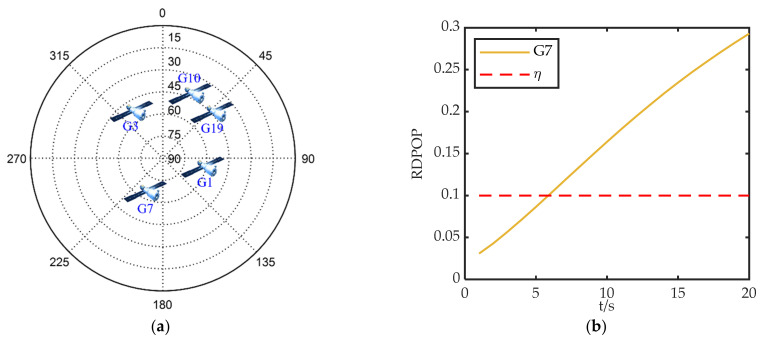
Satellite observation environment: (**a**) the configuration of the satellites; (**b**) RDPOP of faulty satellite.

**Figure 10 sensors-22-05313-f010:**
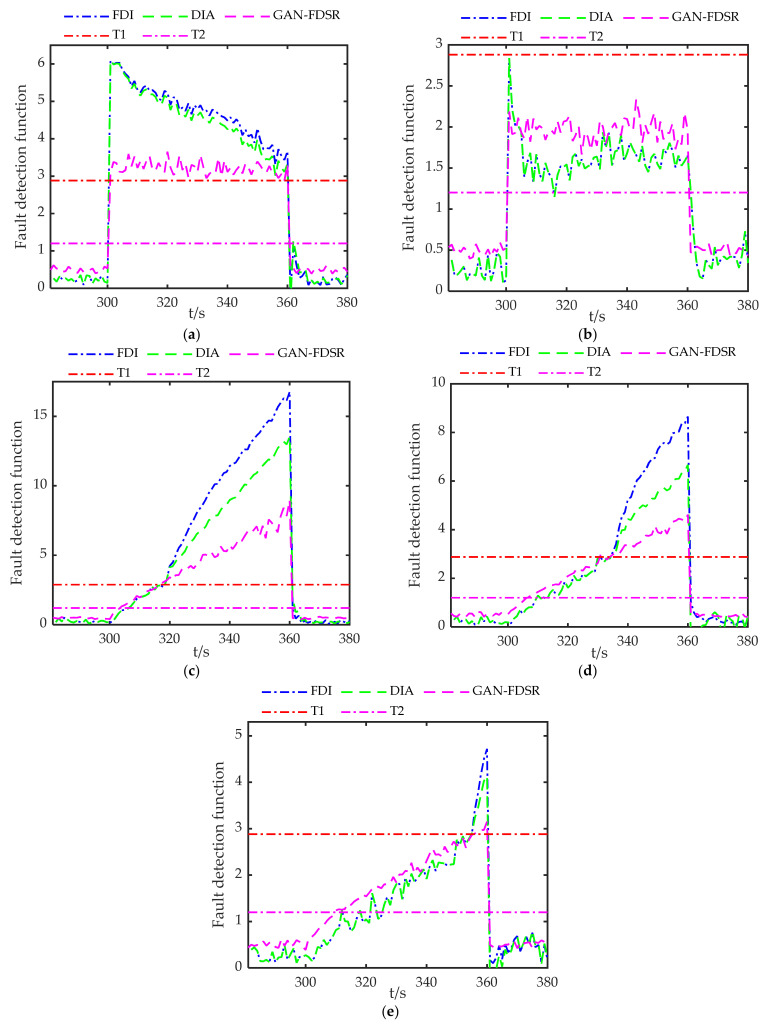
Fault detection functions for three methods: (**a**–**e**) show the fault detection functions under Type 1, Type 2, Type 3, Type 4 and Type 5 faults, respectively.

**Figure 11 sensors-22-05313-f011:**
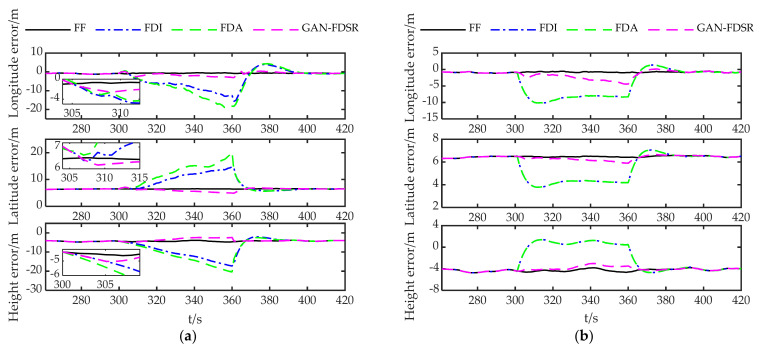
Positioning errors for three methods: (**a**–**e**) show the positioning errors under Type 1, Type 2, Type 3, Type 4 and Type 5 faults, respectively.

**Figure 12 sensors-22-05313-f012:**
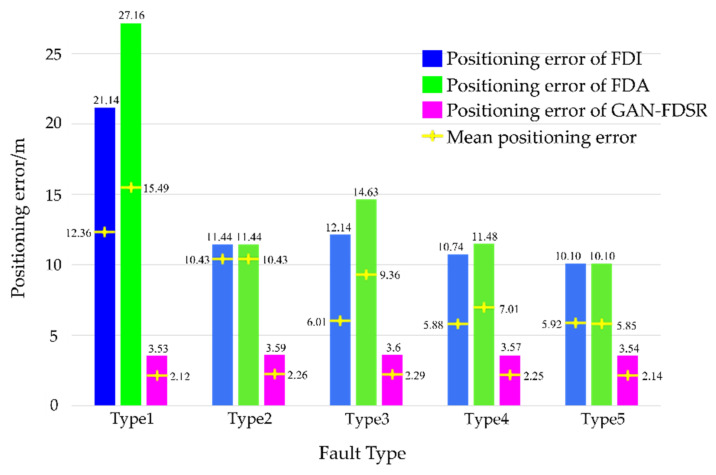
Positioning errors relative to a fault-free system.

**Table 1 sensors-22-05313-t001:** Chaos indices of the visible satellites.

Satellites	τ	m	Largest Lyapunov Exponent
G1 ^1^	8	7	0.2105
G3	7	11	0.1081
G6	8	11	0.1436
G7	6	9	0.2074
G8	10	9	0.1545
G10	7	9	0.1694
G11	6	9	0.1176
G17	7	10	0.1898
G20	5	11	0.2333

^1^ G1: The 1th satellite.

**Table 2 sensors-22-05313-t002:** Inertial sensor technical specifications.

Inertial Sensor	Bias	Bias Stability	Measuring Range
Gyro	0.5°/h	0.5°/h	± 300°/s
Accelerometer	25 mg	25 mg	±10 g

**Table 3 sensors-22-05313-t003:** Satellite observation environments.

	Fault Section	Faulty Satellite	Number of Visible Satellites	Geometric Configuration
Env 1	199–219	G6	9	/
Env 2	199–219	G6	5	/
Env 3	256–276	G11	5	Configuration 1
Env 4	256–276	G11	5	Configuration 2

**Table 4 sensors-22-05313-t004:** Fault types.

Fault Type	Failure Mode	Characteristics of Fault
Type 1	Step fault	20 m
Type 2	Step fault	10 m
Type 3	Gradual fault	1 m/s
Type 4	Gradual fault	0.5 m/s
Type 5	Gradual fault	0.3 m/s

**Table 5 sensors-22-05313-t005:** Performance of fault detection.

	Detection Delay			Missing Alarm Rate		
	FDI	FDA	GAN-FDSR	FDI	FDA	GAN-FDSR
Type 1	0	0	0	0	0	0
Type 2	60	60	0	100%	100%	0
Type 3	18	18	3	30%	30%	5%
Type 4	34	34	6	57%	57%	10%
Type 5	54	54	9	90%	90%	15%

## Data Availability

Not applicable.

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
