# Peer review of "GAN-FDSR: GAN-Based Fault Detection and System Reconfiguration Method"

_sensors, 2022, doi:10.3390/s22145313_

Round 1

Reviewer 1 Report

A GAN-based fault detection and system reconstruction method for tightly coupled GNSS/INS integrated system is proposed in this paper. Overall, the paper is well written and organized with a proper length. The contributions as well as the quality are both good. In addition, there are some points that are not very clear and should be addressed in the revised version:

1.   Please update the references, especially for recent years.

2.   The innovation of this paper is not clear and it is difficult for readers to understand the main contributions of this paper. This part should be added in Introduction section.

3.   The description of the existing work should be shorter in Introduction section.  Furthermore, more descriptions of the proposed method are needed.

4.   The reviewers recommend that more future work should be added on Conclusion Section. For example, incipient fault diagnosis is an important research issue in recent decades. The authors should supplement some results on this aspect, for example the following references had given significant design results:

[1] Incipient winding fault detection and diagnosis for squirrel-cage induction motors equipped on
CRH trains
. ISA Transactions, 2020, 99: 488~495.

Reviewer 2 Report

This paper employs GAN in the fault detection of tightly coupled GNSS/INS integration system and uses different reconfiguration/processing strategies after fault detection with the purpose to improve the integrity and reliability of the system. The proposed method is complicated and many methods such as RDPOP detection are used. The manuscript is not clearly written in the current version. Many symbols are not explained, and equations are not well derived, which can make readers confused. The following is a list of comments which might be of help to improve the quality of the manuscript.

1.       In row 11, “tight coupled systems” should be “tightly coupled systems”.

2.       In row 154, the meaning of “G8” should be explained. The 8th satellite? Should be explained there.

3.       How to determine or get the probabilities used in equation (12)?

4.       What does “CAO” mean in row 172?

5.       What does “phase space” mean in the paper? Please explain it.

6.       In equation (15), what do “X”, “X1”  mean? Are they measurements? If they are, please use other symbols to denote them as X has been used for standing for KF state.

7.       In equation (18), What does “delta_i” mean?.

8.       What does “GRU” mean in row 226?

9.       In equation (22), will the number inside sqrt be negative? And how to ensure that will not happen?

10.   In row 286, what does “S^D” mean? The classification accuracy?

11.   Only a few parameters of gyro are given in Table 2. No accelerometer in POS320?

12.   Are the satellite faults shown in Table 3 and Figure 7 natural or artificial?

13.   KF is used in the integration, but why there is no transient stage in the position errors shown in Figure 8?

14.   What is the threshold “T1” used for in Figure 10?

15.   In the Conclusions, it is stated that the method can reduce the false alarm rate. But no test for false alarm rate is presented in the manuscript.

16.   In Appendix B. Fault Isolation, the calculation of the estimate of Xk is different from the standard KF. Please explain why? Especially, is the expression of equation (A5) reasonable?

17.   In Appendix C. Fault Adaptation, should the vk be Zk in equation (A8) since vk is unknown in practice.

Round 2

Reviewer 2 Report

Each of the comments was addressed and corresponding correction was made in the revised manuscript.  The following minor revisions are suggested to further improve the manuscript.

1. The full name of CAO cannot be found in the manuscript though an explanation is added.

2. Equations (A5) and (A6) should be derived in more detail as they are still not clear.
